# Inter-individual differences in pain anticipation and pain perception in migraine: Neural correlates of migraine frequency and cortisol-to-dehydroepiandrosterone sulfate (DHEA-S) ratio

**Gyöngyi Kökönyei**[1,2,3]*, **Attila Galambos**[2,4], **Natália Kocsel**[2], **Edina Szabó**[2,5], **Andrea Edit Édes**[1], **Kinga Gecse**[1,3], **Dániel Baksa**[1,3], **Dorottya Pap**[3], **Lajos R. Kozák**[6], **György Bagdy**[3,7], **Gabriella Juhász**[1,3]

1 SE-NAP2 Genetic Brain Imaging Migraine Research Group, Hungarian Brain Research Program, Semmelweis University, Budapest, Hungary, 2 Institute of Psychology, ELTE Eötvös Loránd University, Budapest, Hungary, 3 Department of Pharmacodynamics, Faculty of Pharmacy, Semmelweis University, Budapest, Hungary, 4 Doctoral School of Psychology, ELTE Eötvös Loránd University, Budapest, Hungary, 5 Center for Pain and the Brain (PAIN Research Group), Department of Anesthesiology, Critical Care and Pain Medicine, Boston Children's Hospital, Harvard Medical School, Boston, MA, United States of America, 6 MR Research Center, Semmelweis University, Budapest, Hungary, 7 MTA-SE Neuropsychopharmacology and Neurochemistry Research Group, Hungarian Academy of Sciences, Semmelweis University, Budapest, Hungary

* kokonyei.gyongyi@pharma.semmelweis-univ.hu, kokonyei.gyongyi@ppk.elte.hu

**Data Availability Statement:** All data underlying the findings described in the paper, i.e. self-report

## Abstract

Previous studies targeting inter-individual differences in pain processing in migraine mainly focused on the perception of pain. Our main aim was to disentangle pain anticipation and perception using a classical fear conditioning task, and investigate how migraine frequency and pre-scan cortisol-to-dehydroepiandrosterone sulfate (DHEA-S) ratio as an index of neurobiological stress response would relate to neural activation in these two phases. Functional Magnetic Resonance Imaging (fMRI) data of 23 participants (18 females; mean age: 27.61± 5.36) with episodic migraine without aura were analysed. We found that migraine frequency was significantly associated with pain anticipation in brain regions comprising the midcingulate and caudate, whereas pre-scan cortisol-to DHEA-S ratio was related to pain perception in the pre-supplementary motor area (pre-SMA). Both results suggest exaggerated preparatory responses to pain or more general to stressors, which may contribute to the allostatic load caused by stressors and migraine attacks on the brain.

## Introduction

Migraine, as one of the most debilitating diseases [1], is considered to be a complex neurological disorder characterized by subtle changes in functioning of the brain which is related to variations in the structure and connectivity of specific brain areas [2–4]. Altered functioning of

data, raw cortisol and DHEA-S data along with main fMRI contrast maps, are fully available at https://osf.io/g9vph/. Only raw imaging dataset cannot be shared publicly, because at the time our study started, there was no information on open access data availability in the consent forms (the study was approved by the Scientific and Research Ethics Committee of the Medical Research Council (Hungary)), therefore authors are not allowed to share raw imaging data publicly, since participants were not able to accept or refuse their assent to share imaging data in an open access repository. However, raw imaging data are available from the corresponding author (Gyöngyi Kökönyei, kokonyei.gyongyi@ppk.elte.hu) or from the Department of Pharmacodynamics, Faculty of Pharmacy, Semmelweis University (titkarsag. gyhat@pharma.semmelweis-univ.hu) on reasonable request.

**Funding:** The study was supported by the MTA-SE-NAP B Genetic Brain Imaging Migraine Research Group, Hungarian Academy of Sciences, Semmelweis University (Grant No. KTIA_NAP_13-2-2015-0001) to GJ; Hungarian Brain Research Programe (Grant No. 2017-1.2.1-NKP-2017-00002) to GJ and the Hungarian Academy of Sciences (MTA-SE Neuropsychopharmacology and Neurochemistry Research Group) To GB. The preparation of this work was supported by the Thematic Excellence Programme (Tématerületi Kiválósági Program, 2020-4.1.1.-TKP2020) of the Ministry for Innovation and Technology in Hungary, within the framework of the Neurology and Translational Biotechnology thematic programmes of the Semmelweis University. LRK was supported by the Bolyai Research Fellowship Program of the Hungarian Academy of Sciences. The preparation of this article for GK was supported by the Hungarian National Research, Development and Innovation Office (FK128614). GJ was supported by the Hungarian National Research, Development and Innovation Office, Hungary (2019-2.1.7-ERA-NET-2020-00005), under the frame of ERA PerMed (ERAPERMED2019-108). The funders had no role in study design, data collection and analysis, decision to publish, or preparation of the manuscript.

**Competing interests:** The authors have declared that no competing interests exist.

the brain is not limited to migraine attacks (ictal phase) but is present between-attacks (interictal phase) as well. For instance, in fMRI studies, applying acute painful stimuli, migraineurs compared to control subjects had altered pain-induced activation in a broad range of cortical and subcortical structures, including the somatosensory cortices [5, 6], cingulate [5, 7], insula [8], dorsolateral prefrontal cortex (DLPFC) [5, 8], brainstem [9, 10], cerebellum [10], the striatum [11], and the temporal lobe [12].

However, there is accumulating evidence that there are inter-individual differences in pain processing in migraine. Migraine frequency is an important clinical variable. For instance, a recent systematic review [13] shows that headache frequency is an important risk factor for migraine chronification in episodic migraine. With increasing frequency of attacks and years with migraine, the reorganization of grey [14] and white matter [15] and functional connectivity [11] is more and more likely constituting an allostatic load in the long term, since every migraine attack can be considered as a hit on the brain [16]. Thus, it can be considered as a proxy of allostatic load of migraine. In line with this, we can expect that more changes in pain processing occur with more frequent migraine attacks or more years with migraine. Previous results support this notion: for instance, a positive relationship between migraine frequency and activation in several brain areas including the cingulate, DLPFC, fusiform gyrus, precentral gyrus, hippocampus to painful heat stimuli has been detected in Schwedt et al's study [5]. Mathur and colleagues [8] found a positive relationship between activation in bilateral posterior insula to thermal noxious stimuli and migraine frequency, albeit on a lower threshold. BOLD response in the pons to trigeminal painful stimulation correlated positively with attack frequency [6]. Duration of migraine (years with migraine) showed a positive association with increased activation in the fusiform gyrus [5], cerebellum [17], and superior temporal gyrus [8].

It is important to note that pain perception is influenced by expectations and anticipation [18, 19] as illustrious placebo and nocebo studies in this field have showed [20]. Furthermore, in conditioning paradigms, subjective perception and neural response to noxious stimulus are enhanced if it is preceded by cues predicting pain [18], which is in line with the predictive coding model of brain functioning [21]. This means that our pain perception is not simply based on upcoming and ascending sensory input but our previous experiences, and hence related expectations also shape our perception [19]. Taking into account that cues predicting subsequent pain are salient and appraised as threats [22] that help to initiate appropriate actions or adjust behaviour accordingly, it is unsurprising that neural response to pain anticipation have been demonstrated to be widespread, including, for instance, the thalamus, caudate, anterior cingulate cortex, midcingulate and insula [23].

Some evidence suggests that pain anticipation-related neural response in chronic pain compared to pain-free subjects, is enhanced (e.g. in fibromyalgia [24]), however, data about pain anticipation in migraine are scarce, so as its relationship with migraine frequency. For instance, in a placebo study pain anticipation in migraine, compared to control subjects, was associated with enhanced neural response in some regions of the primary visual cortex [25]. Results showing impaired habituation to repeated (and predicted) contact-heat pain in migraine also support the notion that anticipatory processes could be enhanced [26]. In the context of migraine, as attacks become recurrent, pain and the related negative emotional, cognitive or social consequences become more salient, thus any cues associated with pain (as an important symptom of a migraine attack) may have a stronger impact. It is therefore plausible to suggest that there is an association between migraine frequency and the heightened pain anticipation in migraine.

It should be also noted, that not just waiting for pain (or a pain task) could be a stressful situation, but the fMRI procedure itself [27], especially if it is the first time exposure to the fMRI

scanner environment [28]. In line with this, experienced stress before the scan indexed by pre-scan cortisol was associated with enhanced emotional processing of fearful faces relative to neutral ones in limbic structures [29]. However, dehydroepiandrosterone (DHEA) or its sulphated metabolite (dehydroepiandrosterone sulfate; DHEA-S), precursors for adrogens, released during acute stress [30, 31] may counteract the effects of cortisol [32], supported by findings showing that acute administration of DHEA decreased the cortisol level [33]. DHEA and DHEA-S (collectively referred here to DHEA(S)) are also associated with better performance under stressful condition [34]. Besides, accumulating evidence suggests that DHEA(S) is associated with neural processing of emotional stimuli [35]. For instance, exogenous DHEA administration, relative to placebo, reduced the activity of amygdala and hippocampus, and enhanced the activity of medial prefrontal cortex (or more precisely, the rostral ACC) during a task requiring implicit emotional processing and regulation [36].

What is important, based on the opposing effects of cortisol and DHEA(S), that cortisol-to-DHEA(S) ratio has been suggested to be more informative than the two parameters alone [37]. For instance, higher cortisol-to-DHEAS ratio has been found to be associated with all-cause mortality in a prospective cohort [38], and with shorter time till reoccurrence of major depressive episode in a subsample of participants with major depressive disorder in a prospective study [39]. Findings of Grillon and colleagues [40] on the relationship between cortisol-to-DHEA-S ratio and heightened fear potentiated startle in an aversive conditioning task lend support for the notion that cortisol-to-DHEA(S) ratio might be associated with emotion/threat processing, as one of the mechanisms by which complex pattern of adrenocortical activity exerts its effect on health. Thus, it is reasonable to believe that inter-individual differences in pre-scan ratio of cortisol-to-DHEA(S) will be related to pain processing.

The present study therefore aimed to investigate neural response to pain predictive cues and painful stimulus in episodic migraine. More specifically, we were interested in how inter-individual differences in migraine frequency would be related to anticipation and processing of experimentally induced pain between migraine attacks (interictal state). In light of previous studies on pain processing [5, 6, 8, 17], we expected that migraine frequency would be related to pain perception. We further hypothesized that neural anticipatory response to pain predicting cues would also be related to this clinical variable. In the context of pre-scan stress, we considered that cortisol-to-DHEA-S ratio reflects inter-individual differences in stress responsivity to a new and potentially stressful situation, and assumed that this ratio, as an index of neuroendocrine stress reaction, would influence functional activation to pain anticipation and pain perception. In the context of migraine, we supposed that allostatic load exerted by increased anticipatory neuroendocrine stress reaction could contribute to the adverse effects of everyday stressors, resulting changes in brain states. Based on previous evidence on the role of medial prefrontal cortex (mPFC) in stress [41], emotion [42] and pain regulation [43], we expected that higher pre-scan cortisol-to-DHEA-S ratio reflecting increased neuroendocrine stress reaction would enhance pain anticipation and/or perception in mPFC.

## Methods

### Participants

Right-handed subjects aged between 18–38 years were recruited via newspapers, headache clinics and university advertisements. Inclusion criteria were having episodic migraine without aura diagnosed by a headache specialist using the International Classification of Headache Disorders-III criteria (ICHD-III, beta version; Headache Classification Committee of the International Headache Society [44]. The exclusion criteria were the followings: any current or past serious medical, neurological (except migraine), Axis I psychiatric disorders and psychotropic

medication use. Participants were screened using the Mini International Neuropsychiatric Interview [45] by trained researchers. All participants had normal or corrected-to-normal vision. They were invited to the scan if they did not have migraine attacks 48 h prior to the scan, and their data was analysed if they did not have migraine attacks during 24 h follow-up after the scan. They refrained from taking any analgesics 48 h before the scan session and did not take any prophylactic medication during the last three months. Participants were asked to refrain from alcohol for at least 24 hours and from caffeine for at least 4 hours before the scanning session.

Of the 28 participants (22 females, mean age = 27.46, SD = 4.89, age range: 20 to 37 years) taking part in the pain task, five subjects were excluded: three by technical reasons (missing images/volumes originated from technical problems or incomplete voxel level information), and two subjects did not see any correspondence between visual signals and electric stimuli according to the post-task interview (see below). Thus, finally data of 23 subjects (18 females) with migraine without aura (mean age = 27.61, SD = 5.36) were analysed. Mean body mass index (BMI) was 21.52 (SD = 3.72) kg/m$^2$, and all participants had a normal BMI (except one who was obese, his BMI was: 32.9). Nine women were in the follicular phase, and 9 were in the luteal phase. Six women reported using hormonal contraception.

The study was approved by the Scientific and Research Ethics Committee of the Medical Research Council (Hungary, number: 23609-1/2011-EKU, 23421-1/2015-EKU) and written informed consents were received from all subjects in accordance with the Declaration of Helsinki.

## Blood sample collection

Blood samples were collected from participants to 3 ml K3EDTA tubes. The samples were immediately centrifuged, then plasma samples were frozen at -80˚C (-112˚F) until the assay.

Plasma cortisol and dehydroepiandrosterone sulphate (DHEA-S) levels have been measured by competitive ELISA kits from NovaTec Immunodiagnostica GmbH (Dietzenbach, Germany) according to the manufacturer's instructions. Plasma samples were thawed only once. Inter- and intra-assay variance, expressed as coefficient of variation (CV) were: Cortisol: 11.0% and 5.1%; DHEA-S: 10.4% and 7.9%, respectively. Cortisol antibody shows 46.2% cross reaction with prednisolone and DHEA-S antibody shows cross reaction with DHEA (100%) and androstenedione (59%).

DHEA-S was measured in µg/ml and cortisol was measured in ng/ml, their ratio (cortisol-to-DHEA-S) was calculated in µg/ml, and its log-transformed value was used in the analyses based on the recommendation of Sollberger and Ehlert [46]. For descriptive purposes, raw cortisol and DHEA-S values are reported in Table 1. The raw values were unrelated (rho = 0.07, p>0.05).

fMRI scans were administered between 4:00pm and 8:00pm in order to avoid the impact of circadian variations in cortisol [47]. Level of DHEA-S as an important marker of adrenal

**Table 1. Experimental design.**

| Number of trials | Cue | Stimulus |
|---|---|---|
| 10 | Pain cue | Painful (VAS = 7)–Pain |
| 5 | Pain cue | Non-painful (VAS = 3)–Omitted pain |
| 15 | No pain cue | Non-painful (VAS = 3)–Touch |

VAS Visual Analog Scale. Fifteen trials for pain cue and 15 trials for no pain cue.

androgen synthesis also shows a diurnal decline [48], therefore all the participants were scanned in the same time interval.

## Self-report measures

Migraine-related clinical variables: Besides demographic data, all participants were asked to provide information on 1) age at migraine onset; 2) years with migraine (duration of the disease); 3) migraine frequency (average number of migraine per month).

Anxiety: State-Trait Anxiety Inventory (STAI [49]) was used to measure trait- and state-like anxiety symptoms by 20 items. Items are answered on a 4-point Likert-type scale (1 = almost never to 4 = almost always). The internal consistency of both scales was excellent (Cronbach's alpha: 0.84 and 0.93, respectively).

## Psychological task

Two electronic stimuli delivered by Digitimer boxes (Digitimer DS7A, Digitimer Ltd, Welywyn Garden City, UK) were used during the fMRI session that were applied to the dorsum of subject's right hand. The intensity of stimulation was individually set before the session using a staircase protocol. Participants rated the stimuli in a 10-point visual analogue scale (VAS). When the stimulus was rated as 3 (VAS = 3, currents ranging from 0.16 to 0.78 mA) it was set as a non-painful one. Electric shock was augmented to painful but a tolerable stimulus, rated as 7 (VAS = 7, currents ranging from 0.29 to 1.7 mA). Each shock lasts for 2-ms as in Spoormaker and colleagues study [50]. E-Prime script (Psychology Software Tools, Inc., Pittsburgh, PA, USA) controlled the input of Digitimer boxes.

Participants laid still during the scanning session and they had to pay attention to the screen where they could see two different shapes that would be followed by the stimuli. They were not informed about the cue-stimulus relationship to prevent expectation formation based on verbal suggestion. In total, 30 trials in two scanning runs composed the experimental paradigm. The green triangle (15 trials) was always followed by a non-painful stimulus, meanwhile the red square (15 trials) was followed by a painful stimulus. Five trials were exceptional; in these cases, the pain cue was followed by a non-painful stimulus (VAS rating = 3) (omitted pain trials) (see Table 1), as in our previous study [51].

Participants had to report in a post-task interview, if they found any correspondence between cues and subsequent stimuli during the task. During the inter-trial interval, participants saw only a black screen lasting 30 s. In the task, each visual signal was preceded by a white fixation cross for 1 s (see Fig 1). The pseudorandomized visual stimuli duration was ranged from 6 to 12 s (average = 9.3 s).

## fMRI acquisition

In this study, a whole-brain functional MRI data acquisition was carried out at a 3T MRI scanner (Achieva 3T, Philips Medical Systems, Best, The Netherlands) using a BOLD-sensitive T2*-weighted echo-planar imaging sequence (repetition time [TR] = 2,500 ms, echo time [TE] = 30 ms, field of view [FOV] = $240 \times 240$ mm2) with 3 mm $\times$ 3 mm in-plane resolution and contiguous 3-mm slices providing whole-brain coverage. A series of high-resolution anatomical images were also obtained with a T1-weighted 3D TFE sequence with $1 \times 1 \times 1$ mm resolution.

## Statistical analysis of self-report data

To analyse demographic and self-report data, t-tests and correlation analyses were used with two-tailed $p < 0.05$ threshold in SPSS version 25.0 (IBMSPSS, IBMCorp, Armonk, NY, USA).

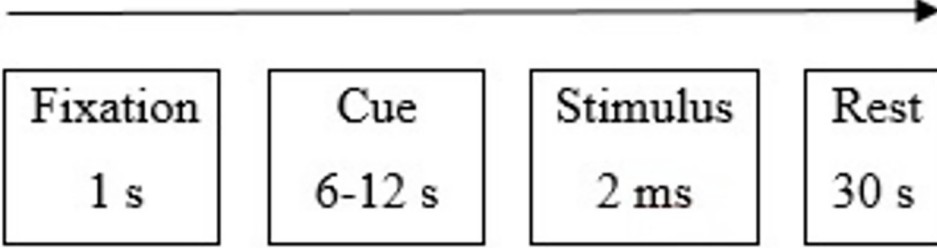

**Fig 1. Design of the pain task.**

## fMRI data analysis

Regarding preprocessing and first-level model we followed the same procedure described in our previous study [51].

**Preprocessing.** Imaging data analysis was carried out using the Statistical Parametrical Mapping (SPM12) analysis software package (Wellcome Department of Imaging Neuroscience, Institute of Neurology, London, UK; http://www.fil.ion.ucl.ac.uk/spm12/) implemented in Matlab 2015b (Math Works, Natick, MA, USA). Preprocessing of the functional images contained realignment, co-registration to the structural image, segmentation, normalization in Montreal Neurological Institute (MNI) space, and spatial smoothing with an 8-mm full width half-maximum Gaussian kernel. In the last step of pre-processing, quality of images was visually inspected.

**First-level model.** First level fMRI models included the fixation cross, each of the two cues (pain cue and no-pain cue) and the three outcomes (touch, pain, omitted pain; see Table 5). The effects of low-frequency physiological noise were removed using high-pass temporal filtering with a cut-off of 128 s. An autoregressive AR (1) model was used to serial correlations in data series. We identified the motion outliers (threshold of global signal > 3 SD and motion > 1mm) with the Artifact Detection Tools (ART www.nitrc.org/projects/artifact_detect/) and along with the six motion parameters they were used as regressors of no interest in the first level fMRI model.

First-level analysis was performed on each participant to determine the significant BOLD signal responses to anticipation of pain (pain cue vs. no pain cue) and to perception of pain (painful stimuli (VAS = 7) vs. non-painful one (VAS = 3).

**Second level analysis.** The above described contrast maps were then entered into the second-level analyses. First, one sample t-tests were performed to examine anticipation-related activations in the whole brain. Another whole-brain activation analysis was conducted to identify activations in the pain perception period.

In the second step, migraine frequency as a clinical variable was used as a covariate in the analyses to reveal the associations between BOLD responses to pain cue and painful stimuli. Based on previous studies (see the introduction) we tested whether migraine frequency correlated with BOLD signal in the pain perception period, and then we explored its relationship with pain anticipation (see introduction) using whole-brain regression analyses. Then we investigated whether the log transformed cortisol-to-DHEA-S ratio associated with BOLD response in the medial PFC, using the mask of de la Vega and co-workers [52] to test our a-priori hypothesis. We lacked cortisol-to-DHEA-S ratio for one subject who was excluded from this analysis.

Considering group-level whole-brain analysis, we used a nonparametric, permutation-based approach (SnPM13; http://www.nisox.org/Software/SnPM13/) with 5000 permutations

and without variance smoothing in order to control for false-positives due to multiple testing [53]. Whole-brain analyses were carried out at a p<0.001 uncorrected level and cluster-level family-wise error-corrected pFWE <0.05 values were reported as significant. Activated clusters were identified anatomically using the automated anatomical labelling atlas [54]. For ROI analyses, we performed small volume correction (SVC) analyses in SPM using voxel-wise threshold p<0.05, FWE corrected.

Trait anxiety has been demonstrated to be associated with anticipation of pain [55], and state anxiety seems to affect perceived pain intensity [56] and sensitivity [57] to experimental pain, therefore we also checked whether results changed if trait or state anxiety was added as a covariate.

We used a gray matter mask in our analyses, the template we used was provided by the Brain Imaging Centre, Montreal Neurological Institute, McGill University: https://digital.lib.washington.edu/researchworks/handle/1773/33312).

Statistical maps were visualized on the MNI 152 template brain provided in MRIcroGL (http://www.mccauslandcenter.sc.edu/mricrogl/) [58]

## Results

Clinical characteristics and demographic data of participants along with descriptive data of psychological scales are shown in Table 2. Trait and state anxiety correlated positively (rho = 0.49, p<0.05), but anxiety scores were unrelated to age, education level and clinical characteristics (correlational coefficients were between -0.12 and 0.15, p>0.05). Endocrine parameters, i.e. raw cortisol and DHEAS were not related to age, anxiety scores, clinical variables, menstrual phase and hormonal contraception taking. Raw cortisol and cortisol-to-DHEA-S ratio correlated negatively with BMI (rho = -0.43 p<0.05 and rho = -0.55 p<0.01, respectively). Correlational results are reported in the S1 and S2 Tables.

### fMRI results

**Task-related activations.** *Pain anticipation.* Increased activation to pain cues versus no pain cues was elicited in bilateral occipital cortex (calcarine, lingual gyrus), bilateral middle

**Table 2. Demographic data and clinical characteristics of participants (N = 23).**

|  | Frequency (%) or Mean (*SD*) | Range |
|---|---|---|
| Women | 18 (78.3%) |  |
| Age | 27.61 (5.36) | 20–37 |
| Highest education |  |  |
| High school | 11 (47.8%) |  |
| Graduate degree | 12 (52.2%) |  |
| Age at migraine onset | 15.59 (7.16) | 6–29 |
| Number of years with migraine | 12.02 (7.97) | 2–30 |
| Migraine frequency/month | 2.35 (2.40) | 1–12 |
| State anxiety | 34.26 (8.59) | 23–67 |
| Trait anxiety | 39.04 (7.87) | 27–54 |
| Cortisol (ng/ml) | 136.33 (90.00) | 8.52–319.11 |
| DHEA-S (μg/ml) | 1.92 (1.01) | 0.41–5.22 |
| Cortisol/DHEA-S (μg/ml) | 0.09 (0.9) | 0.003–0.412 |

Note. DHEAS: dehydroepiandrosterone sulphate. Cortisol, DHEAS and cortisol-to-DHEAS ratio for one subject were missing.

**Table 3. Activation changes during pain anticipation (N = 23).**

| Contrast | Cluster size (voxels) | Region | Br | Side | Peak T-value | MNI coordinates | | |
|---|---|---|---|---|---|---|---|---|
| | | | | | | x | y | z |
| Pain cue > No pain cue | 246 | Calcarine | | L | 5.68 | -9 | -88 | 11 |
| | | Lingual gyrus | | L | 5.28 | -9 | -76 | -1 |
| | | Calcarine | 18 | R | 5.12 | 6 | -85 | 11 |
| | 87 | Midcingulate | | R | 4.23 | 12 | 14 | 41 |
| | | Midcingulate | 24 | L | 4.09 | -6 | 2 | 44 |
| | | SMA | 6 | R | 4.04 | 6 | 2 | 56 |
| No pain cue > Pain cue | 75 | Inferior occipital gyrus | | L | 5.22 | -42 | -73 | -7 |
| | | Inferior occipital gyrus | | L | 5.03 | -42 | -61 | -13 |
| | 83 | Superior parietal lobule | | R | 5.13 | 33 | -61 | 53 |
| | | Superior occipital gyrus | | R | 4.43 | 27 | -61 | 32 |

Note: cluster-level familywise error rate of p<0.05; R: right; L: left; SMA: Supplementary motor area.

cingulate gyrus and right supplementary motor area (see Table 3 and Fig 2), while decreased activation in inferior occipital gyrus and superior parietal lobule was detected.

Results did not change substantially when gender, trait or state anxiety were controlled for (except in one case: when gender was controlled for, the analysis for the no pain cue vs. pain cue contrast resulted only one significant cluster in the inferior occipital gyrus (k = 69); these results are reported in the S3–S5 Tables).

*Pain perception.* Contrasting painful (VAS = 7) versus non-painful stimuli (VAS = 3) led to widespread activation in brain areas including the cerebellum, vermis, insula, postcentral gyrus, supramarginal gyrus, middle temporal gyrus, Rolandic operculum and Heschl gyrus (see Table 4 and Fig 2). These result did not change substantially when state anxiety or gender

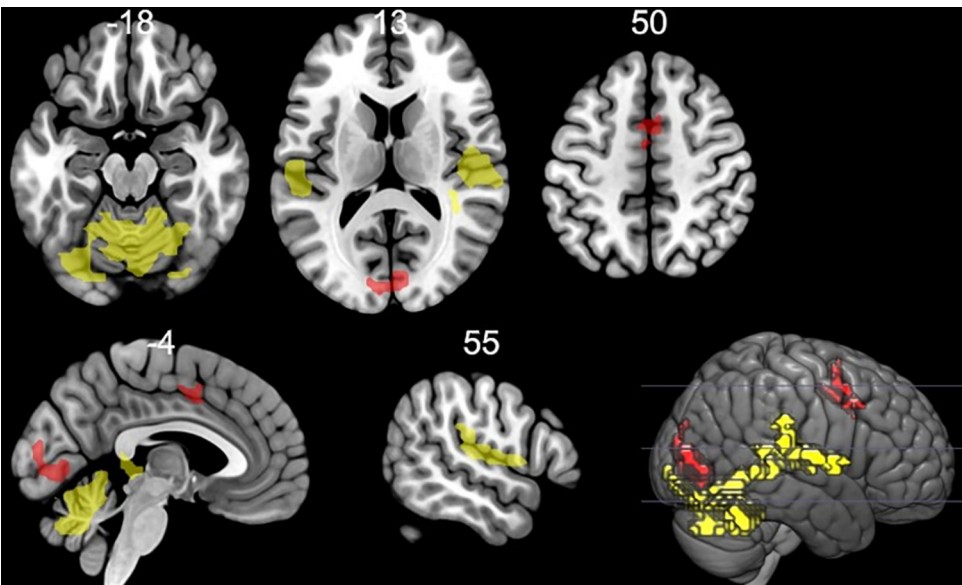

**Fig 2. Significant activations to pain anticipation (pain cue vs. no pain cue—red) and perception (painful vs. non-painful stimulus—yellow).** Significant activated clusters are shown at pFWE < 0.05. Coordinates are in Montreal Neurological Institute (MNI) space. Statistical maps were visualized on the MNI 152 template brain provided in MRIcroGL (http://www.mccauslandcenter.sc.edu/mricrogl/) [58].

Table 4. Activation changes to painful and non-painful stimulation (N = 23).

| Contrast | Cluster size (voxels) | Region | Side | Peak T-value | MNI coordinates | | |
|---|---|---|---|---|---|---|---|
| | | | | | x | y | z |
| Pain > No pain | 1138 | Vermis 8 | L | 7.00 | -3 | -64 | -28 |
| | | Cerebellum_6 | R | 6.03 | 33 | -49 | -31 |
| | | Cerebellum_6 | R | 6.00 | 30 | -55 | -25 |
| | 206 | Supramarginal gyrus | L | 5.59 | -63 | -28 | 29 |
| | | Postcentral gyrus | L | 5.50 | -57 | -19 | 17 |
| | | Insula | L | 4.94 | -39 | -16 | 8 |
| | 366 | Middle temporal gyrus | R | 5.37 | 45 | -43 | -1 |
| | | Rolandic operculum | R | 5.01 | 60 | -1 | 5 |
| | | Heschl gyrus | R | 4.84 | 42 | -22 | 11 |

Note: cluster-level familywise error rate of p<0.05. R: right; L: left.

was entered as a control variable in the analysis (see S6 and S7 Tables). The other contrast (non-painful stimuli (VAS = 3) vs. painful stimuli (VAS = 7)) did not produce any significant voxels.

*Regression analyses with migraine frequency and cortisol/DHEAS ratio*. Migraine frequency was correlated with activation in the caudate, and midcingulate to pain cue vs. no-pain cue (see Table 5 and Fig 3), but was unrelated to pain perception (this analysis did not produce any significant voxels).

Using the mask of mPFC by de la Vega and coworkers [52] we tested whether pre-scan cortisol-to-DHEA-S ratio related to pain anticipation and perception. We found that cortisol-to-DHEA-S ratio was associated with activation in the right supplementary motor area (peak MNI coordinates x/y/z = 3/14/50; t(1, 20) = 5.32 SVC, pFWE = 0.041) to painful vs. non-painful stimuli positively, but was unrelated to pain anticipation (this analysis did not produce any significant voxels). Since the cortisol-to-DHEA-S ratio correlated negatively with BMI, we repeated our analyses while controlling for BMI, but the results were unchanged.

## Discussion

Previous fMRI studies on pain processing in migraine focused primarily on pain perception, demonstrating an altered response to acute painful stimuli and its relationship with some clinical variables (e.g. migraine frequency). In our study we aimed to disentangle anticipation and perception processes of painful stimuli and examine how inter-individual differences in migraine frequency and pre-scan neurobiological stress response would relate to these two phases.

Table 5. Relationship between migraine frequency (average number of migraine attacks per month) and pain anticipation.

| Contrast | Cluster size (voxels) | Region | Br | Side | Peak T-value | MNI coordinates | | |
|---|---|---|---|---|---|---|---|---|
| | | | | | | x | y | z |
| Pain cue–No pain cue | 586 | caudate | | R | 5.16 | 15 | 17 | 2 |
| | | midcingulate | 24 | R | 4.84 | 3 | 17 | 32 |
| | | caudate | | L | 4.27 | 21 | 17 | 17 |

Note: Cluster-level familywise error rate of p<0.05. R: right; L: left; NA: coordinates are not in the AAL.

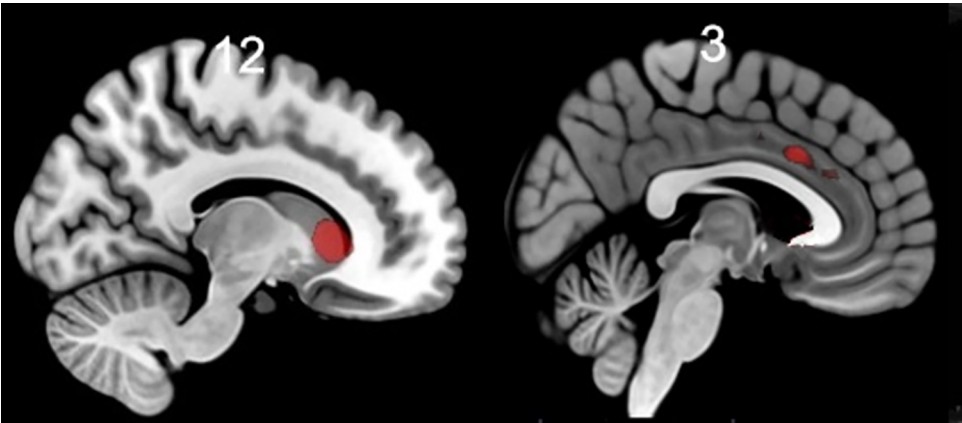

**Fig 3. The association between pain anticipation and migraine frequency.** Significantly activated cluster (caudate on the left and midcingulate on the right) is shown at pFWE < 0.05. Statistical map was visualized on the MNI 152 template brain provided in MRIcroGL (http://www.mccauslandcenter.sc.edu/mricrogl/) [58]. Coordinates are in Montreal Neurological Institute (MNI) space.

Repeated (or more frequent) attacks have been considered as a potent migraine-related factor leading to allostatic load [16, 59]. Regarding pre-scan neurobiological stress response, the cortisol-to-DHEAS ratio to the first time exposure to fMRI scanner environment was calculated. We considered that this ratio reflects inter-individual differences in stress responsivity to a new and potentially stressful situation. Thus, the neural correlates of both indices might help to elucidate how or why allostatic load induced by repeated attacks or stressors contribute to changes in brain states. Using a classical fear conditioning task, we found that migraine frequency was associated with enhanced pain anticipation, whereas pre-scan neurobiological stress response was related to pain perception. We start our discussion with pain perception since most of the studies investigated this process.

## Pain perception

Generally, several parts of the pain processing brain network [60] were activated. For instance, we found that the perception of painful stimuli vs. non-painful one was associated with widespread activation in the posterior part of the cerebellum and vermis, with peaks in the cerebellar lobule VI and posterior vermis (VIII lobule) in episodic migraine without aura. This elevated activity is in line with previous studies demonstrating cerebellar activation to painful stimuli [12, 61]. More specifically, functional connectivity of the cerebellum with the brainstem and higher cortical areas such as the insula during painful stimuli [61] suggest a modulatory role of cerebellum in pain processing [12]. In addition, overlapping activation in the cerebellum (including the lobule VI, Crus I and VIIb) to noxious heat and aversive pictures was found, leading to the notion that the cerebellum may play a role in encoding aversive stimuli in general [12]. In the context of migraine without aura, cerebellar activity to trigeminal nociceptive stimulation [6] along with microstructural alterations in the posterior part of the cerebellum [10, 62] supports the notion that cerebellum may contribute to the pathophysiology of migraine.

Next to cerebellar activity, we also found increased activity in two clusters: one cluster involving the left posterior insula, postcentral gyrus (including the primary somatosensory cortex, S1 Table) and supramarginal gyrus, and the other involving the Rolandic operculum, middle temporal gyrus and Heschl gyrus. According to a recent meta-analysis by Xu and colleagues [63] the insula, supramarginal gyrus and Rolandic operculum–along with the

thalamus, secondary somatosensory cortex and MCC–are consistently activated to painful stimuli irrespective of the experimental paradigm used in healthy participants.

In our study, the insular activity to painful stimuli was located in the posterior subdivision which is considered to reflect the sensory-discriminative properties of pain [64]. The functional and structural connectivity of posterior insula (PI) with secondary somatosensory cortex further supports its involvement in sensory-discriminative processing [65]. Regarding the timing of pain response in the insula, it occurs first in the PI [66], then the anterior parts are also activated [66, 67] which is considered to play a role in the encoding of the emotional aspects of pain. In the present study we applied a short-duration pain and did not assess subjective experience of the painful stimuli on a trial-by-trial basis. This may explain why we did not find any activation in the anterior insula [68].

Due to the lack of a control group, we could only investigate within-group activations to painful stimulation, but the activated areas in our study have relevance in between-group comparisons as well, as previous studies show. Among female migraineurs without aura, altered structural and functional properties of the PI has been documented [69]. For instance, altered tactile sensory processing indexed by reduced habituation in posterior insula along with enhanced activation in brainstem to repeated blocks of innocuous ophthalmic nerve somatosensory stimulation was found in episodic migraine [70]. As a hub region of the brain, insula plays a role in information processing derived from the external or internal milieu, thus it has been proposed as one of the key regions in migraine pathophysiology [2] since altered sensory processing has been commonly reported in migraine [4].

Altered structural and functional properties of primary somatosensory cortex has also been detected in migraine [71]. Comparing groups, activation of S1 Table to noxious thermal stimuli was higher among high-frequency migraine compared to low-frequency migraine [11]. In addition, activation in S1 Table to fearful vs. neutral face also showed a positive relationship with migraine frequency [72]. Interestingly, in both mentioned studies the activation corresponded with head/face area of the sensory homunculus.

Contrary to previous findings, in our study migraine frequency did not relate to the perception of painful (threatening) stimulus. It might be due to methodological issues, since with our task we could disentangle anticipation and perception periods, while tasks in previous studies were not designed such a way.

**Pain perception and pre-scan stress.** Increased anabolic imbalance reflected in cortisol-to-DHEA-S ratio to a new stressor may reflect individual vulnerability, therefore we expected that inter-individual differences in pre-scan stress reflected in higher cortisol-to-DHEA-S ratio would be associated with the perception of painful stimuli. We restricted our analysis to the mPFC since it is a common site for stress [41], emotion [42] and pain regulation [43]. We found that pre-scan cortisol-to-DHEAS ratio was associated with activation in the right supplementary motor area to painful stimulus vs. non-painful one. Thus, generally, our finding supports the notion that cortisol-to-DHEA(S) ratio might be related to threat processing.

Activation of SMA has been demonstrated in many pain studies [68, 73], and although it is traditionally a motor area, its engagement in pain processing is often interpreted as reflecting motor/behaviour planning in response to pain [74]. Based on this, our result might indicate that those with higher cortisol-to-DHEA-S ratio are more ready to produce behavioural responses to painful stimuli (e.g. escape from the stimuli). The site of the activation corresponds with pre-SMA [75], which is presumably involved not just in motor responses, but in more complex processes, such as response inhibition [76] as well. Thus alternatively, our results might suggest that those who experience higher pre-scan stress indicated by higher cortisol-to-DHEA(S) ratio have to employ more resources to inhibit (automatic) motor responses to painful stimuli.

We did not find any significant associations between state anxiety and pre-scan cortisol level, similar to previous studies [77]. However, the relationship between them may be dependent on the type of the stressor. For instance, cortisol level before a sport competition and state anxiety correlated positively [78].

## Pain anticipation

Given its aversive and salient nature, pain elicits fear, as do pain predictive cues [79] along with widespread brain activation [23]. However, previous studies on pain processing in migraine have rarely investigated anticipation processes. In the present study, the activation of occipital–visual–areas corresponds with our previous result using the same task [51], and might be due to the long duration of anticipatory cues [80].

In addition, contrasting pain cues versus no pain cues, we found activations in the mid-cingulate and SMA. Activation on both areas has been detected in the anticipation phase of pain [81], and in one study activation in SMA, for instance, correlated with the expected pain intensity positively [82]. In addition, a recent meta-analytic study found that MCC is part of the core network involved in aversive anticipation [80]. Increased reactivity in the MCC to aversive anticipatory cues in fMRI studies may reflect that this area has a role in preparing for the aversive stimulation and/or initiating avoidance or defensive reactions [83]. However, this activation is presumably not specific to pain or pain cues [84, 85], as the MCC, especially its anterior part, is involved in orienting to the emotional salience of information. In line with this, cortical hyperresponsivity towards sensory information in migraine [4] might be associated with greater anticipatory responses in the MCC. Though we could not test this directly, since we did not have a control group. However, it is worth mentioning that functional connectivity of the anterior MCC was associated with clinical variables–such as attack frequency, disease duration and pain intensity–in a previous study in migraine [86].

Amplified anticipation response is not rare in pain conditions (e.g. in fibromyalgia), or other neurological diseases (e.g. Parkinson disease [87]). Our findings highlight that investigating anticipation processes, and hence expectations, can be fruitful in migraine as well.

**Pain anticipation and migraine frequency.** We found that migraine frequency was associated with increased neural response to pain anticipation in a relatively large cluster involving the MCC along with the caudate. As mentioned in the previous section, increased reactivity in the MCC to aversive anticipatory cues in fMRI studies may reflect that this area has a role in preparing for the aversive stimulation and/or initiating avoidance or defensive reactions [83], but it also has a role in evaluating the salience of stimuli. Based on functional connectivity studies, the ventral caudate is also considered to be involved in affective processing [88]. Our results correspond with this notion, since anticipation of noxious stimuli (i.e. pain) has an indisputable biological significance, allowing preparation in responses to upcoming noxious stimuli [89].

It is worth mentioning, that the role of basal ganglia in pathophysiology of migraine has been proposed [90], supported by functional and structural alterations among migraineurs [11]. However, it is difficult to determine whether it is a correlate for progression of the illness, or it is a pre-morbid vulnerability factor.

Based on our findings, we may conclude that more frequent attacks are accompanied with increased preparatory responses to painful—or in more general threatening—stimuli, establishing another path for repeated headaches and stressors to exert their effects on brain in migraine.

## Limitation

There are some limitations to our study. First, we did not have a control group, so we could not test between-group differences. However, our main aim was to investigate how inter-individual differences in migraine frequency and pre-scan stress are associated with pain anticipation and perception. Regarding migraine-related variables, we chose migraine frequency, since it has been associated with brain alteration [11, 14, 15] and chronification of episodic migraine [13]. However, we cannot exclude the possibility that migraine-related variables other than migraine frequency had an association with anticipatory processes. For instance, previous findings showed that monetary loss anticipation was associated with the time of the last attack [91]. In addition, migraine cycle can also alter pain processing. For instance, brainstem activation was modulated by migraine cycle, resulting activation differences in inter-ictal, pre-ictal and post-ictal groups [9]. In our study, we scanned our participants in the inter-ictal phase, thus subsequent studies should provide additional data on pain anticipation across migraine cycle.

We did not measure cutaneous allodynia reflecting central sensitization processes, which may have important implications for pain processing. For instance, migraine-related allodynia is associated with resting state functional connectivity of areas involved in descending pain modulation (e.g. PAG) [92]. Interestingly, Schwedt and colleagues [93] found that based on structural brain characteristics, there are two subgroups of patients with migraine, differing on allodynia, duration of disease and disability scores, pointing out that identification of subgroups on meaningful characteristics is warranted to provide more tailored therapies or interventions.

We used electrical stimulation to trigger pain, which activates not just nociceptive, but non-nociceptive afferents as well. But according to a recent EEG study, similarly to laser evoked stimulation, electric simulation is also suitable to investigate expectation-related pain processing [94]. However, future studies using painful stimulus in different pain modalities as well as non-painful aversive stimuli will help to characterize anticipatory processes in migraine. Regarding our pain task, other methodological issues could be also relevant. For instance, unlike many studies [18, 95] we did not ask our participants to rate the stimuli delivered to their hand, and a long period of rest was used, which might affect our results. Due to the task parameters (i.e. pain stimuli occurred immediately after the offset of cues), we could not test how predictions (anticipatory cues) affect pain perception. Though it is known that threat prediction can bias pain perception [18, 19], it would be interesting to see whether migraineurs are more biased towards prediction than towards nociceptive inputs.

Detection of mismatch between the anticipated/expected and experienced stimuli generates prediction error signals that makes possible to the precision of our expectation based on the incoming signals/inputs. In the context of pain, the activity of anterior insula, especially its ventral part [96], using computational modelling, was best explained if the prediction error component was included in the model [97]. In tasks using omitted pain vs. pain contrast representing receiving lower-than-expected pain, recruitment of the ACC, especially its rostral part was found [98]. Though we had five trials in which pain cue was followed by non-painful stimulus, called omitted pain trials, we did not analyse omitted pain vs. pain contrast, since we think that this contrast would deserve between-group comparison.

We did not have the possibility to see how cortisol, DHEA-S or their ratio changed from pre-scan to post-scan, however it would have been informative even for pain processing. For instance, in one study, those who had the largest reactive cortisol response to the pain task rated the pain less unpleasant, and showed smaller activity in several areas including the posterior insula, primary somatosensory cortex, anterior MCC, and nucleus accumbens [99].

Last, but not least our sample size may limit the generalizability of our findings, and might contribute to the lack of common brain activations to pain stimulus in the thalamus and ACC [63, 100].

## Conclusion

Based on our finding that migraine frequency is related to enhanced anticipation in areas involved in preparatory response, we may conclude that this is another path through which repeated headaches and/or stressors may create an allostatic load on the brain and the body in migraine. Additionally, expectations about upcoming migraine attacks, or generally, stressors in daily life, may have important implications for quality of life in migraine. In line with this, further studies should focus on characterizing anticipatory processes and expectations in migraine with different methods, and their possible involvement in chronification of migraine with prospective studies. Our results may also highlight that decreasing migraine frequency either pharmacologically or with other methods (e.g. with psychological treatment [101]) could be important to minimize allostatic load resulting from frequent attacks and related phenomena. Importantly, our neuroimaging findings on pre-scan biological stress response and pain perception also underlines preparatory responses in migraine. Overall, exaggerated preparedness on a biological and/or a behavioural level may contribute to the allostatic load [102] caused by stressors and migraine attacks on the brain in migraine [16].

## Supporting information

**S1 Table. Spearman correlations of anxiety scores with age, education level and clinical characteristics (N = 23).** *p<0.05; [1]point biserial correlation.
(DOCX)

**S2 Table. Spearman correlations of pre-scan raw cortisol, DHEAS and cortisol-to-DHEAS with age, anxiety scores, clinical variables, menstrual phase and hormonal contraception taking (N = 23).** *p<0.05; [1]point biserial correlation.
(DOCX)

**S3 Table. Activation changes during pain anticipation if state anxiety is controlled for (N = 23).** Cluster-level familywise error rate of p<0.05; R, right; L, left; SMA, Supplementary motor area.
(DOCX)

**S4 Table. Activation changes during pain anticipation if trait anxiety is controlled for (N = 23).** Cluster-level familywise error rate of p<0.05; R, right; L, left; SMA, Supplementary motor area.
(DOCX)

**S5 Table. Activation changes during pain anticipation if gender is controlled for (N = 23).** Cluster-level familywise error rate of p<0.05; R, right; L, left; SMA, Supplementary motor area.
(DOCX)

**S6 Table. Activation changes to painful and non-painful stimulation if state anxiety was controlled for (N = 23).** Cluster-level familywise error rate of p<0.05; R, right; L, left.
(DOCX)

**S7 Table. Activation changes to painful and non-painful stimulation if gender was controlled for (N = 23).** Cluster-level familywise error rate of p<0.05; R, right; L, left.
(DOCX)

**S8 Table. Activation changes during pain anticipation controlled for gender, state and trait anxiety (N = 23).** Cluster-level familywise error rate of p<0.05; R, right; L, left; SMA, Supplementary motor area.
(DOCX)

**S9 Table. Activation changes to painful and non-painful stimulation controlled for gender, state and trait anxiety (N = 23).** Cluster-level familywise error rate of p<0.05; R, right; L, left.
(DOCX)

## Acknowledgments

The fMRI study was conducted as a groupwork at MR Research Center, Semmelweis University. The authors thank Ádám György Békésy-Szabó, Krisztina Oláh Koósné, István Kóbor, Márk Folyovich, Krisztina Kovács, Zita Nagy, Zsuzsanna Tóth, Terézia Zsombók, Máté Magyar and Éva Csépány for their contribution and József Edőcs for the assistance of programming Digitimer boxes.

## Author Contributions

**Conceptualization:** Gyöngyi Kökönyei, Gabriella Juhász.

**Data curation:** Gyöngyi Kökönyei.

**Formal analysis:** Gyöngyi Kökönyei, Attila Galambos.

**Funding acquisition:** György Bagdy, Gabriella Juhász.

**Investigation:** Attila Galambos, Natália Kocsel, Edina Szabó, Andrea Edit Édes, Dorottya Pap.

**Methodology:** Gyöngyi Kökönyei, Kinga Gecse, Dániel Baksa, Lajos R. Kozák, Gabriella Juhász.

**Project administration:** Edina Szabó, Dorottya Pap.

**Resources:** Lajos R. Kozák.

**Visualization:** Gyöngyi Kökönyei, Andrea Edit Édes.

**Writing – original draft:** Gyöngyi Kökönyei, Kinga Gecse.

**Writing – review & editing:** Gyöngyi Kökönyei, Attila Galambos, Natália Kocsel, Edina Szabó, Andrea Edit Édes, Kinga Gecse, Dániel Baksa, Dorottya Pap, Lajos R. Kozák, György Bagdy, Gabriella Juhász.

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
