## [Decision Letter · Decision Letter 0]

6 Oct 2021

PONE-D-21-27604Inter-individual differences in pain anticipation and pain perception in migraine: neural correlates of migraine frequency and cortisol-to-dehydroepiandrosterone sulfate (DHEA-S) ratioPLOS ONE

Dear Dr. Gyöngyi Kökönyei,

Thank you for submitting your manuscript to PLOS ONE. After careful consideration, we feel that it has merit but does not fully meet PLOS ONE’s publication criteria as it currently stands. Therefore, we invite you to submit a revised version of the manuscript that addresses the points raised during the review process.

We look forward to receiving your revised manuscript.

Kind regards,

Mukesh Dhamala, Ph. D.

Academic Editor

PLOS ONE

Journal Requirements:

Additional Editor Comments:

As you can see, the manuscript was reviewed by one expert and was assessed for a minor revision. We ask the authors to address the critiques of the reviewer and revise the manuscript accordingly.

Reviewers' comments:

Reviewer's Responses to Questions

**Comments to the Author**

1. Is the manuscript technically sound, and do the data support the conclusions?

Reviewer #1: Yes

2. Has the statistical analysis been performed appropriately and rigorously? 

Reviewer #1: Yes

3. Have the authors made all data underlying the findings in their manuscript fully available?

Reviewer #1: No

4. Is the manuscript presented in an intelligible fashion and written in standard English?

Reviewer #1: Yes

5. Review Comments to the Author

Reviewer #1: Summary: In this manuscript, authors used fear conditioning fMRI task to study the association of migraine frequency and neurobiological stress response with neural activation in two specific phases i.e., pain anticipation and pain perception. Authors found that migraine frequency was associated with pain anticipation in a cluster comprising the midcingulate and caudate, whereas pre-scan neurobiological stress response was related to pain perception in the pre-supplementary motor area (pre-SMA). Overall, this is an interesting study with very nicely written Introduction and Discussion sections, detailed Methods, and well-organized Results. Biggest limitation is the small sample size of only 23 participants. I have few major and mostly minor concerns and suggestions related to this manuscript as pointed below:

Abstract:

- It would make more sense to define DHEA-S as an index of neurobiological stress response at the start of the abstract when it was first mentioned rather in the results.

- It’s not clear what do the authors mean by “one cluster” comprising the midcingulate and caudate – given that midcingulate is a cortical whereas caudate is a sub-cortical region. It’s not clear how both these regions lead to one comprised cluster?

- “Pre-supplementary motor area (pre-SMA) region” should be changed to just “pre-supplementary motor area (pre-SMA)”.

- Last sentence needs revision – “migraine attacks on the brain in migraine” is confusing.

Keywords:

- I would avoid using the keywords that are already present in the title. Instead, authors should use other keywords to broaden the search criteria of their paper.

Introduction:

- Introduction is very well written with detailed overview of literature and very clear hypotheses and predictions.

Methods:

- Line 158: “Right-handed subjects with migraine between 18–38 years” needs revision. It appears the subjects had migraine between 18 and 38, rather I would suggest mentioning sample size, mean and SD of age, age range, and male/female details here – for the full initial sample.

- Line 167: “three by technical reasons” should be “three due to technical reasons”, and any brief details about the technical reasons would be helpful.

- Line 171: Authors mentioned that participants did not have migraine attacks 48 h prior and 24 h after the scan. Was this by chance or this was one of the inclusion criteria, and for 24 h after the scan, did the authors follow-up with participants? These details are missing here.

- Lines 296-297: “Whole-brain analysis…” needs revision.

Results:

- Line 351: It’s confusing when authors mention three clusters but then report eight clusters. Similarly, lines 364, 369, 422-425, and Table 5. This issue needs to be fixed throughout the manuscript.

Discussion: Very detailed and nicely written.

General comments/questions and suggestions:

- Why did the authors prefer to report Spearman correlations rather than Pearson’s?

- Did the authors perform quality check on outliers while performing correlation analysis?

- Authors report supplementary data with state anxiety, trait anxiety, and gender as controlled variables but individually. Why didn’t author use all these three variables together as a set of covariates instead of using those one-by-one?

- Suggestion for authors for their future work is to use this data and explore if structural data (e.g., morphometry parameters such as cortical/sub-cortical volume) and brain connectivity measures (e.g., directed functional and effective) have any association with migraine frequency, stress response parameters and other behavioral data, if available, from this project.

6. PLOS authors have the option to publish the peer review history of their article (what does this mean?). If published, this will include your full peer review and any attached files.

Reviewer #1: No

---

## [Author Response · Author response to Decision Letter 0]

28 Nov 2021

Dear Dr. Mukesh Dhamala,

We thank you that you are willing to consider a revision. We would like to take the opportunity and resubmit the revised version of our manuscript, entitled “Inter-individual differences in pain anticipation and pain perception in migraine: neural correlates of migraine frequency and cortisol-to-dehydroepiandrosterone sulfate (DHEA-S) ratio”.

Regarding data availability, we would like to complete our previous statement. We can share the self-report data and raw cortisol and DHEA-S values along with main main fMRI contrast maps, however raw imaging data cannot be shared publicly, but request can be sent to our department as well. “Self-report data, raw cortisol and DHEA-S data along with main fMRI contrast maps are available at https://osf.io/g9vph/. Raw imaging dataset cannot be shared publicly, because at the time our study started, there was no information on open access data availability in the consent forms (the study was approved by the Scientific and Research Ethics Committee of the Medical Research Council (Hungary)), therefore authors are not allowed to share raw imaging data publicly, since participants were not able to accept or refuse their assent to share imaging data in an open access repository. However, raw imaging data are available from the corresponding author (Gyöngyi Kökönyei, kokonyei.gyongyi@ppk.elte.hu) or from the Department of Pharmacodynamics, Faculty of Pharmacy, Semmelweis University (titkarsag.gyhat@pharma.semmelweis-univ.hu) on reasonable request.” 

We were asked to review our reference list. It is complete and correct.

We thank the reviewer for their feedback, comments and suggestions. Please find below our detailed answers to the reviewer’s comments (we attached our responses in a separate file as well). In the revised manuscript the original text is presented in black, modifications are indicated by tracked changes mode in MS Word. We hope that we have managed to address each of the reviewer’s concerns.

Sincerely,

the authors

Reviewer #1: Summary: In this manuscript, authors used fear conditioning fMRI task to study the association of migraine frequency and neurobiological stress response with neural activation in two specific phases i.e., pain anticipation and pain perception. Authors found that migraine frequency was associated with pain anticipation in a cluster comprising the midcingulate and caudate, whereas pre-scan neurobiological stress response was related to pain perception in the pre-supplementary motor area (pre-SMA). Overall, this is an interesting study with very nicely written Introduction and Discussion sections, detailed Methods, and well-organized Results. Biggest limitation is the small sample size of only 23 participants. I have few major and mostly minor concerns and suggestions related to this manuscript as pointed below:

Abstract:

- It would make more sense to define DHEA-S as an index of neurobiological stress response at the start of the abstract when it was first mentioned rather in the results.

#Response: Thank you for taking the time to review our manuscript! We changed the abstract accordingly. In the text: “Previous studies targeting inter-individual differences in pain processing in migraine mainly focused on the perception of pain. Our main aim was to disentangle pain anticipation and perception using a classical fear conditioning task, and investigate how migraine frequency and pre-scan cortisol-to-dehydroepiandrosterone sulfate (DHEA-S) ratio as an index of neurobiological stress response would relate to neural activation in these two phases.”

- It’s not clear what do the authors mean by “one cluster” comprising the midcingulate and caudate – given that midcingulate is a cortical whereas caudate is a sub-cortical region. It’s not clear how both these regions lead to one comprised cluster?

#Response: Using cluster-extent based thresholding can results clusters that cover multiple anatomical brain regions. It is the case for our analysis when we tested correlates of migraine frequency in pain anticipation. 

We set primary threshold to p<0.001 as suggested by Woo et al. (2014, Neuroimage) and used non-parametric, permutation-based approach (SnPM13; http://www.nisox.org/Software/SnPM13/) with 5000 permutations and without variance smoothing in order to control for false-positives due to multiple testing, and cluster-level family-wise error-corrected pFWE <0.05 values were reported as significant. 

According to Woo and colleagues (2014) a primary threshold of p<0.001 is considered to “be sufficient to identify regions that are localized enough to be anatomically interpretable in many studies”, and especially recommended for studies with low or moderate sample size (N<50). In addition, we chose non-parametric, permutation-based approach to determine the minimum cluster-extend threshold to prevent that false positives are greatly inflated. Thus, we think that the mentioned cluster though cover regions that are spatially distinct is interpretable.

However, to avoid misunderstanding we rephrased this sentence in the abstract:

“We found that migraine frequency was significantly associated with pain anticipation in brain regions comprising the midcingulate and caudate, whereas pre-scan cortisol-to DHEA-S ratio was related to pain perception in the pre-supplementary motor area (pre-SMA).”

- “Pre-supplementary motor area (pre-SMA) region” should be changed to just “pre-supplementary motor area (pre-SMA)”.

#Response: Thank you, we changed the abstract accordingly.

- Last sentence needs revision – “migraine attacks on the brain in migraine” is confusing.

#response: We deleted “in migraine” from the sentence. 

Keywords:

- I would avoid using the keywords that are already present in the title. Instead, authors should use other keywords to broaden the search criteria of their paper.

#Response: Thank you very much for the suggestion, we changed our keywords to the following: headache; fMRI; stress response; pre-SMA; caudate

Introduction:

- Introduction is very well written with detailed overview of literature and very clear hypotheses and predictions.

#Response: Thank you very much for the feedback. 

Methods:

- Line 158: “Right-handed subjects with migraine between 18–38 years” needs revision. It appears the subjects had migraine between 18 and 38, rather I would suggest mentioning sample size, mean and SD of age, age range, and male/female details here – for the full initial sample.

#Response: Thank you, we re-wrote the first sentence of the Participants section: “Right-handed subjects aged between 18-38 years were recruited …”

In addition, we added all the important details for the full initial sample. In the text: “Of the 28 participants (22 females, mean age=27.46 , SD=4.89, age range: 20 to 37 years) taking part in the pain task, … “

- Line 167: “three by technical reasons” should be “three due to technical reasons”, and any brief details about the technical reasons would be helpful.

#Response: We corrected the sentence, and added details about the technical reasons. In the text: “…five subjects were excluded: three due to technical reasons (missing images/volumes originated from technical problems or incomplete voxel level information), and two subjects did not see any correspondence between visual signals and electric stimuli according to the post-task interview (see below).”

- Line 171: Authors mentioned that participants did not have migraine attacks 48 h prior and 24 h after the scan. Was this by chance or this was one of the inclusion criteria, and for 24 h after the scan, did the authors follow-up with participants? These details are missing here.

#Response: It was an inclusion criteria not to have migraine attacks 48 h prior the scan, or more precisely, we scheduled the scans accordingly, and the data were analysed if participants did not have attacks 24 h after the scan, thus we followed-up our participants. To make it clear, we moved up these details to the paragraph about inclusion criteria and added these pieces of information to the text. 

In the text: “They were invited to the scan if they did not have migraine attacks 48 h prior to the scan, and their data was analysed if they did not have migraine attacks during the 24 h follow up after the scan. They refrained from taking any analgesics 48 h before the scan session and did not take any prophylactic medication during the last three months. Participants were asked to refrain from alcohol for at least 24 hours and from caffeine for at least 4 hours before the scanning session.”

- Lines 296-297: “Whole-brain analysis…” needs revision.

#Response: Thank you, we corrected it. 

Results:

- Line 351: It’s confusing when authors mention three clusters but then report eight clusters. Similarly, lines 364, 369, 422-425, and Table 5. This issue needs to be fixed throughout the manuscript.

#Response: The text can be confusing, however it is correct. We identified only 3 statistically significant clusters, but all the clusters contained more regions (see our explanation above). For instance in Table 5 three clusters with 8 regions are reported. However, to avoid misunderstanding we rephrased this sentence:

“Contrasting painful (VAS=7) versus non-painful stimuli (VAS=3) led to widespread activation in brain areas including the cerebellum, vermis, insula, postcentral gyrus, supramarginal gyrus, middle temporal gyrus, Rolandic operculum and Heschl gyrus (see Table 4 and Figure 2)."

Discussion: Very detailed and nicely written.

#Response: Thank you very much for the feedback.

General comments/questions and suggestions:

- Why did the authors prefer to report Spearman correlations rather than Pearson’s?

#Response: Since some of the variables (e.g. state anxiety) were not normally distributed, therefore we decided to use Spearman correlations.

- Did the authors perform quality check on outliers while performing correlation analysis?

#Response: thank you for the question. When we calculated correlational coefficients we checked whether there are any outliers using Simple Scatterplot in SPSS, and if any outliers were detected we checked whether results changed after excluding them. We did not find substantial differences in these cases, so we came to the conclusion that outliers were not influential.

- Authors report supplementary data with state anxiety, trait anxiety, and gender as controlled variables but individually. Why didn’t author use all these three variables together as a set of covariates instead of using those one-by-one?

#Response: We decided to use them one-by-one in our analyses in the first place since our sample size was small. However, based on your comment, we checked whether results would change if all the three variables were used as a set of covariates. You can see the results below which were unchanged. We added these tables to the Supporting information file. (Please see S8 and S9 Tables, and we also added these tables to the Response to Reviewers file).

- Suggestion for authors for their future work is to use this data and explore if structural data (e.g., morphometry parameters such as cortical/sub-cortical volume) and brain connectivity measures (e.g., directed functional and effective) have any association with migraine frequency, stress response parameters and other behavioral data, if available, from this project.

#Response: Thank you very much for the suggestion, we are currently working on additional analyses as you also suggested. 

We also thank you for the constructive critics and the useful suggestions.

---

## [Editor Report · Decision Letter 1]

6 Dec 2021

Inter-individual differences in pain anticipation and pain perception in migraine: neural correlates of migraine frequency and cortisol-to-dehydroepiandrosterone sulfate (DHEA-S) ratio

PONE-D-21-27604R1

Dear Dr. Kökönyei,

We’re pleased to inform you that your manuscript has been judged scientifically suitable for publication and will be formally accepted for publication once it meets all outstanding technical requirements.

Kind regards,

Mukesh Dhamala, Ph. D.

Academic Editor

PLOS ONE

---

## [Editor Report · Acceptance letter]

10 Dec 2021

PONE-D-21-27604R1 

Inter-individual differences in pain anticipation and pain perception in migraine: neural correlates of migraine frequency and cortisol-to-dehydroepiandrosterone sulfate (DHEA-S) ratio 

Dear Dr. Kökönyei:

I'm pleased to inform you that your manuscript has been deemed suitable for publication in PLOS ONE. Congratulations! Your manuscript is now with our production department. 

Kind regards, 

on behalf of

Dr. Mukesh Dhamala 

Academic Editor

PLOS ONE